# An Analytic Orthotropic Heat Conduction Model for the Stretchable Network Heaters

**DOI:** 10.3390/mi13071133

**Published:** 2022-07-18

**Authors:** Zeqing He, Yingli Shi, Jin Nan, Zhigang Shen, Taihua Zhang, Zhao Zhao

**Affiliations:** 1Aerospace Information Research Institute, Chinese Academy of Sciences, Beijing 100094, China; zqhe@aoe.ac.cn (Z.H.); zth@aoe.ac.cn (T.Z.); 2School of Materials and Energy, University of Electronic Science and Technology of China (USETC), Chengdu 610054, China; 3Institute of Solid Mechanics, Beihang University (BUAA), Beijing 100191, China; nanjincn@163.com; 4Beijing Key Laboratory for Powder Technology Research and Development, Beihang University (BUAA), Beijing 100191, China; shenzhg@buaa.edu.cn; 5China Special Equipment Inspection and Research Institute, Beijing 100029, China

**Keywords:** orthotropic substrate, heat conduction, stretchable network heater, horseshoe lattice, uniform temperature distribution

## Abstract

Compared with other physiotherapy devices, epidermal electronic systems (EES) used in medical applications such as hyperthermia have obvious advantages of conformal attachment, lightness and high efficiency. The stretchable flexible electrode is an indispensable component. The structurally designed flexible inorganic stretchable electrode has the advantage of stable electrical properties under tensile deformation and has received enough attention. However, the space between the patterned electrodes introduced to ensure the tensile properties will inevitably lead to the uneven temperature distribution of the thermotherapy electrodes and degrade the effect of thermotherapy. It is of great practical value to study the temperature uniformity of the stretchable patterned electrode. In order to improve the uniformity of temperature distribution in the heat transfer system with stretchable electrodes, a temperature distribution manipulation strategy for orthotropic substrates is proposed in this paper. A theoretical model of the orthotropic heat transfer system based on the horseshoe-shaped mesh electrode is established. Combined with finite element analysis, the effect of the orthotropic substrate on the uniformity of temperature distribution in three types of heat source heat transfer systems is studied based on this model. The influence of the thermal conductivity ratio in different directions on the temperature distribution is studied parametrically, which will help to guide the design and fabrication of the stretchable electrode that can produce a uniform temperature distribution.

## 1. Introduction

Recent advancements in flexible and stretchable electronics technology provide a feasible solution for bionic electronic skin [1,2,3,4,5] and bio-integrated electronics [6,7,8]. Specifically, the intrinsic rigid material is thinned and designed as a serpentine ribbon, so that it can withstand large structural deformation with small strain, which can significantly improve the stretchability of inorganic electronics [9,10]. Among the numerous achievements, epidermal electronic systems (EES), as a wearable device with a paradigm shift, is a representative research direction, due to its advantages in thickness, mechanical stiffness and density comparable to human epidermis [11,12]. Flexible wearable heaters, as an important component of EES, play an irreplaceably important role in thermotherapy medicine application [13], such as cutaneous wound healing [14,15], subcutaneous tumor treatment [16], and drug release [17].

The research achievements on stretchable heaters can be divided into two effective strategies. One is to use inherently flexible materials to fabricate stretchable heaters, such as hyper-elastic polymer filled with metal nanowire [18,19], organic conductive polymers [20,21], graphene [22], etc. The limitation of this strategy is that the electrical resistance of this heater will increase rapidly under stretching conditions due to the slippage of the nano-fibers, which will lead to the degrading performance of the heater and the mismatch between the temperature distribution and the design. Another approach is to encapsulate the patterned periodic metal network or liquid metal micro-channel into elastic substrates as stretchable electrodes [23,24,25]. The electrical resistance of this flexible electrode will not drift obviously due to flexible deformation. The flexible electrode design requires a certain spacing to maintain the extensibility [26,27], which will inevitably lead to the inhomogeneous temperature distribution. However, in some application scenarios, the uniformity of temperature distribution of hyperthermia heaters is a very important indicator. The significance of the temperature distribution uniformity provided by the biophysical therapy heaters in EES is that if the temperature distribution produced by the heater is not uniform, it is difficult to choose the appropriate heating power. Local high temperature will damage epidermal tissue, and low temperature cannot produce the effect of heat therapy [26]. If the uniformity of temperature distribution of the hyperthermia electrode is enhanced by reducing the electrode size, when the line width is small to a certain extent, it will not only increase the technological difficulty of micro-nano manufacturing but also reduce the structure robustness of the heating electrode. Referring to the thermal management of orthotropic materials [28,29], the flexible substrate and encapsulation with orthotropic thermal conduction can be utilized to manipulate the heat flux and reconfigure the uneven temperature distribution generated by the stretchable heater. Vemuri et al. [28] proposed a novel design of alternating layered distribution of two materials, which can change the direction of heat flow. Li et al. [30,31] applied the design method of this bi-layer alternating laid thermal metamaterial to the thermal management of flexible μ-ILEDs (micro-scale inorganic light-emitting diodes) attached to human skin. Shi et al. [29] proposed a stretchable composite metamaterial with a periodic layered lattice structure whose anisotropic thermal conductivity can be regulated via external strain. The orthotropic heat conduction substrate has low requirements for the micro-nano process and good thermal management effect, so it is a suitable choice to enhance the temperature distribution uniformity of a hyperthermia electrode.

Here, for the previously proposed wearable network heater based on a negative Poisson’s ratio horseshoe lattice structure [32], we investigate the heat transfer characteristics of its integration with an orthotropic thermal conduction substrate in order to point out its characteristics of manipulating heat flux and homogenization of temperature distribution. Compared to the simple configuration heat sources in previous studies, the analytical orthotropic heat conduction model with the complex horseshoe lattice heat source is developed in this work. The thermal field of the model is calculated using Fourier cosine transform and linear superposition of a horseshoe heat source in Section 2, which is validated by finite element analysis (FEA) in Section 3. Meanwhile, the effects of different geometric parameters of horseshoe heat source (number of sides) and ratios of thermal conductivity in different directions of the substrate on the thermal properties are discussed. Section 4 presents the conclusion.

## 2. Analytical Modeling 

Figure 1a shows that the triangular lattice structure as a network heater embedded between the substrate and encapsulation. Due to periodicity, the thermal field of a single cell can represent that of the whole structure, as shown in Figure 1b. The network heat source can be modeled as a line heat source when its cross-sectional dimension is much smaller than the in-plane dimension. Therefore, the thermal field of the point heat source in the model is shown in Figure 1c, where the temperature distribution of the heater can be obtained by integrating the point heat source along the curve of the line heat source.

The temperature increase from the ambient temperature *T_a_* is denoted by Δ*T_i_* =*T_i_ − T_a_*. The temperature increase in the orthotropic model satisfies the Fourier heat conduction equation as
(1)$$k_{i}^{x}\frac{\partial^{2}\Delta T_{i}}{\partial x^{2}}+k_{i}^{y}\frac{\partial^{2}\Delta T_{i}}{\partial y^{2}}+k_{i}^{z}\frac{\partial^{2}\Delta T_{i}}{\partial z^{2}}=0,$$
where *i* = 1 denotes the encapsulation and *i* = 2 is for the substrate. The *k^x^*, *k^y^* and *k^z^* denote the thermal conductivity in the *x*, *y* and *z*-direction, respectively. The natural convection condition at the upper and lower surfaces yields
(2)$${k_{1}^{z}\frac{\partial \Delta T_{1}}{\partial z}|}_{z=-z_{1}}=h_{0}\Delta T_{1},$$
(3)$${-k_{2}^{z}\frac{\partial \Delta T_{2}}{\partial z}|}_{z=z_{2}}=h_{0}\Delta T_{2},$$
where *h*_0_ is the coefficient of heat convection with air. The continuity of temperature increase and heat flux between the substrate and the encapsulation layer can be obtained by
(4)$${\Delta T_{1}|}_{z=0^{-}}={\Delta T_{2}|}_{z=0^{+}},$$
(5)$${k_{1}^{z}\frac{\partial \Delta T_{1}}{\partial z}|}_{0^{-}}-{k_{2}^{z}\frac{\partial \Delta T_{2}}{\partial z}|}_{0^{+}}=\{\begin{array}{l} Q_{0}=\underset{a,b\to 0^{+}}{\lim}\frac{P}{4ab},\;(x,y)\in D\\ 0,(x,y)\notin D\;\; \end{array},$$
where *Q*_0_ and *P* are the heat flux and the power of a point heat source. The rectangular heat source with an infinitely small side length (*a*, *b*) is equivalent to a point heat source. *D* denotes the region of the heat source. Based on the Fourier cosine transform
(6)$$\Delta {\tilde{T}}_{i}(\alpha ,\beta ,z)=\int_{0}^{\infty }\int_{0}^{\infty }\Delta T_{i}(x,y,z)\cos(\alpha x)\cos(\beta y)dxdy,$$
the heat conduction equations in Equation (1) are converted to ordinary differential equations as
(7)$$\frac{d^{2}\Delta {\tilde{T}}_{i}}{dz^{2}}-\left(\frac{\alpha^{2}k_{i}^{x}+\beta^{2}k_{i}^{y}}{k_{i}^{z}}\right)\Delta {\tilde{T}}_{i}=0.$$

Additionally, the boundary and continuity conditions in Equations (2)–(5) can be written as
(8)$${k_{1}^{z}\frac{d\Delta {\tilde{T}}_{1}}{dz}|}_{z=-z_{1}}=h_{0}\Delta {\tilde{T}}_{1},$$
(9)$${-k_{2}^{z}\frac{d\Delta {\tilde{T}}_{2}}{dz}|}_{z=z_{2}}=h_{0}\Delta {\tilde{T}}_{2},$$
(10)$${\Delta {\tilde{T}}_{1}|}_{z=0^{-}}={\Delta {\tilde{T}}_{2}|}_{z=0^{+}},$$
(11)$${k_{1}^{z}\frac{d\Delta {\tilde{T}}_{1}}{dz}|}_{z=0^{-}}-{k_{2}^{z}\frac{d\Delta {\tilde{T}}_{2}}{dz}|}_{z=0^{+}}=\underset{a,b\to 0}{\lim}\frac{P\sin(\alpha a)\sin(\beta b)}{4ab\alpha \beta }=\frac{P}{4},$$

The general solution for Equation (7) with the boundary conditions and the continuity conditions can be obtained by
(12)$$\Delta {\tilde{T}}_{i}=A_{i}e^{\sqrt{\frac{\alpha^{2}k_{i}^{x}+\beta^{2}k_{i}^{y}}{k_{i}^{z}}}z}+B_{i}e^{-\sqrt{\frac{\alpha^{2}k_{i}^{x}+\beta^{2}k_{i}^{y}}{k_{i}^{z}}}z}.$$

Substituting Equation (12) into Equations (8)–(11), the four coefficients *A*_1_, *A*_2_, *B*_1_ and *B*_2_ can be solved as
(13)$$\begin{array}{l} A_{1}^{}=\frac{p}{4}\frac{\left(h_{0}+C_{1}\right)\left(\left(h_{0}+C_{2}\right)e^{2z_{2}\xi_{2}}+C_{2}-h_{0}\right)}{\left[\begin{array}{l} \left(\left(C_{1}+C_{2}\right)\left(h_{0}+C_{1}\right)-\left(C_{2}-C_{1}\right)\left(h_{0}-C_{1}\right)e^{-2z_{1}\xi_{1}}\right)\left(h_{0}+C_{2}\right)e^{2z_{2}\xi_{2}}\\ -\left(\left(C_{1}+C_{2}\right)\left(h_{0}-C_{1}\right)e^{-2z_{1}\xi_{1}}-\left(C_{2}-C_{1}\right)\left(h_{0}+C_{1}\right)\right)\left(h_{0}-C_{2}\right) \end{array}\right]}\\ A_{2}=\frac{p}{4}\frac{\left(h_{0}-C_{2}\right)\left(\left(h_{0}-C_{1}\right)e^{-2z_{1}\xi_{1}}-C_{1}-h_{0}\right)}{\left[\begin{array}{l} \left(\left(C_{2}-C_{1}\right)\left(h_{0}+C_{2}\right)e^{2z_{2}\xi_{2}}+\left(C_{1}+C_{2}\right)\left(h_{0}-C_{2}\right)\right)\left(C_{1}-h_{0}\right)e^{-2z_{1}\xi_{1}}\\ +\left(\left(C_{1}+C_{2}\right)\left(h_{0}+C_{2}\right)e^{2z_{2}\xi_{2}}+\left(C_{2}-C_{1}\right)\left(h_{0}-C_{2}\right)\right)\left(h_{0}+C_{1}\right) \end{array}\right]}\\ B_{1}^{}=-\frac{A_{1}^{}\left(h_{0}-C_{1}\right)e^{-2z_{1}\xi_{1}}}{h_{0}+C_{1}}\;\\ B_{2}^{}=-\frac{A_{2}^{}\left(h_{0}+C_{2}\right)e^{2z_{2}\xi_{2}}}{h_{0}-C_{2}} \end{array}$$
where
(14)$$\xi_{i}=\sqrt{\frac{\alpha^{2}k_{i}^{x}+\beta^{2}k_{i}^{y}}{k_{i}^{z}}},C_{1}=\xi_{1}k_{1}^{z},C_{2}=\xi_{2}k_{2}^{z},$$

According to the inverse Fourier cosine transform, the temperature increase in the point heat source can be calculated by
(15)$$\Delta T_{i}\left(x,y,z\right)=\frac{4}{\pi^{2}}\int_{0}^{\infty }\int_{0}^{\infty }\Delta {\tilde{T}}_{i}\left(\alpha ,\beta ,z\right)\cos(\alpha x)\cos(\beta y)d\alpha d\beta .$$

The point heat source is integrated along with the network heat source, and the temperature increase in the unit cell is obtained by linear superposition of the surrounding periodic heat sources, which gives
(16)$$\theta_{i}\left(x,y,z\right)=\sum_{n=1}^{\infty }\int_{l_{0}}\Delta T_{n}\left(x,y,z\right)dl,$$
where *l*_0_ denotes the function of the unit cell of the network line heat source. 

## 3. Results and Discussion

In this section, the orthotropic heat conduction model with three kinds of network heat sources (i.e., triangular, square and honeycomb) are investigated and verified by FEA. Take *χ*_0_ as the included angle of two adjacent edges in the stretchable heat source lattice, as shown in Figure 2a; then, the included angles of triangle, quadrilateral and honeycomb mesh heaters are *χ*_0_ = 60°, *χ*_0_ = 90°and *χ*_0_ = 120°, respectively. The functions of different unit cells with triangular, square and honeycomb heat sources can be written as
(17)$$l_{triangular}=\{\begin{array}{c} \begin{array}{c} {\left(x-1\right)}^{2}+y^{2}=1,\left(y\geq 0\right)\\ {\left(x-\frac{1}{2}\right)}^{2}+{\left(y-\frac{\sqrt{3}}{2}\right)}^{2}=1,\left(y-\sqrt{3}x\geq 0\right)\\ {\left(x+\frac{1}{2}\right)}^{2}+{\left(y-\frac{\sqrt{3}}{2}\right)}^{2}=1,\left(y+\sqrt{3}x\leq 0\right) \end{array}\\ {\left(x+1\right)}^{2}+y^{2}=1,\left(y\leq 0\right)\\ {\left(x+\frac{1}{2}\right)}^{2}+{\left(y+\frac{\sqrt{3}}{2}\right)}^{2}=1,\left(y-\sqrt{3}x\leq 0\right)\\ {\left(x-\frac{1}{2}\right)}^{2}+{\left(y+\frac{\sqrt{3}}{2}\right)}^{2}=1,\left(y+\sqrt{3}x\geq 0\right) \end{array}\;l_{square}=\{\begin{array}{c} \begin{array}{c} \begin{array}{c} {\left(x-1\right)}^{2}+y^{2}=1,\left(y\geq 0\right)\\ x^{2}+{\left(y-1\right)}^{2}=1,\left(x\leq 0\right) \end{array}\\ {\left(x+1\right)}^{2}+y^{2}=1,\left(y\leq 0\right) \end{array}\\ x^{2}+{\left(y+1\right)}^{2}=1,\left(x\geq 0\right) \end{array}\;l_{honeycomb}=\{\begin{array}{c} \begin{array}{c} {\left(x-1\right)}^{2}+y^{2}=1,\left(y\leq 0\right)\\ {\left(x-\frac{5}{2}\right)}^{2}+{\left(y-\frac{\sqrt{3}}{2}\right)}^{2}=1,\left(y-\sqrt{3}\left(x-2\right)\geq 0\right)\\ {\left(x-\frac{5}{2}\right)}^{2}+{\left(y+\frac{\sqrt{3}}{2}\right)}^{2}=1,\left(y+\sqrt{3}\left(x-2\right)\geq 0\right) \end{array}\\ {\left(x+1\right)}^{2}+y^{2}=1,\left(y\geq 0\right)\\ {\left(x+\frac{5}{2}\right)}^{2}+{\left(y-\frac{\sqrt{3}}{2}\right)}^{2}=1,\left(y+\sqrt{3}\left(x+2\right)\leq 0\right)\\ {\left(x+\frac{5}{2}\right)}^{2}+{\left(y+\frac{\sqrt{3}}{2}\right)}^{2}=1,\left(y-\sqrt{3}\left(x+2\right)\leq 0\right) \end{array}$$

The stretchable network heat source is made of copper (width: 1 μm, thickness: 0.01 μm, radius: 1 mm and cross-sectional area: 1 × 10^−8^ mm^2^), which is located between the substrate and encapsulation with the thickness of 1 mm, respectively. The triangular network heat source is a regular hexagon with a side length of 2.31 mm, as shown in Figure 2a. The encapsulation and substrate are selected as Ecoflex (e.g., Ecoflex00-10, Smooth-On, Inc., Macungie, PA, USA) with a thermal conductivity of 0.16 W·m^−1^·K^−1^ [29]. The natural air convection coefficient is set as *h* = 15 W·m^−2^·K^−1^. Based on the ratio of current density shown in Figure 2b, the powers of the line heat sources in the horizontal and oblique paths are set as 1.96 × 10^−3^ W and 0.49 × 10^−3^ W, respectively, and the corresponding heat flux densities are 0.3125 W/m and 0.0781 W/m, respectively. The reason is that for the network heat source with the voltage applied at the left and right ends, the ratio of current density in the horizontal and oblique path is 2:1 from the equivalent circuit approach combined with the periodic conditions (see Appendix A, for details). The heat flux density ratio in the horizontal and oblique path should be 4:1 according to Joule’s law. A 3D thermal model is established in ABAQUS. The heater is modeled as a surface heat source. The DC3D20 element is used to discretize the geometry. The element size ranges from 0.01 to 0.05 mm with a total number of over 200,000. The fine mesh is adopted for the region near the heat source and the coarse mesh for that far from the heat source. The convergence of FEA has been verified by the tiny change of less than 0.1% via the comparison with that using a much smaller element size of 0.005 mm (see Appendix A, for details).

As biophysical therapy heaters in EES, the temperature distribution at the substrate/skin interface (i.e., the bottom surface of the substrate) is most noteworthy of the EES heat transfer system, because the temperature and uniformity here play a key role in the effect of physiotherapy. So, we focus on the temperature on the bottom surface of the substrate. Figure 2c shows the comparison between analytical and FEA results of temperature distribution at the bottom of the isotropic and orthotropic substrate with a triangular network heater (*χ*_0_ = 60°), respectively. For the isotropic substrate, the thermal conductivity in three directions is *k_s_^x^* = *k_s_^y^* = *k_s_^z^* = 0.16 W·m^−1^·K^−1^. It can be seen that the temperature distribution of the substrate is not uniform, and the temperature difference in the *y*-direction in a lattice reaches around 5 °C. Because of the voltage applied to the two ends in the *x*-direction of the network heater (Figure 2b), different current densities in the *x*-direction and *y*-direction cause different heat flux densities, resulting in the uneven temperature distribution. In order to improve the uniformity of temperature distribution at the bottom of the substrate, the orthotropic substrate is utilized to ameliorate the heat transfer system; that is, by increasing the thermal conductivity in the *y*-direction (*k_s_^y^* in Equation (1)) to 1.6 W·m^−1^·K^−1^, other parameters remain unchanged. Figure 2c shows that this orthotropic substrate strategy can significantly improve the uniformity of temperature distribution, and the maximum temperature difference of temperature distribution does not appear to be higher than 2 °C, which is verified by FEA.

In order to quantitatively analyze the temperature distribution uniformity of the isotropic substrate and the orthotropic substrate strategy effect on the uniformity optimization of temperature distribution, the temperature distribution along two paths of *x* = 0 and *y* = 0 of two different substrates (the origin is at the geometric center of the network heater lattice) is drawn in Figure 3. It can be observed that along the *y*-direction (*x* = 0) of an isotropic substrate, the temperature rises from 6.3 °C at the lattice edge to 9.6 °C at the center point, and the temperature difference is 3.3 °C, which is calculated in Figure 3a. For the orthotropic substrate, the temperature difference in the *y*-direction (*x* = 0) is only 1.2 °C compared with the isotropic substrate, and the temperature evolves from 7.0 to 8.2 °C along the path. This is because the increase in thermal conductivity *k_s_^y^* helps to obtain a more uniform temperature distribution along the *y*-direction. Figure 3b shows the temperature distribution of the isotropic substrate and orthotropic substrate in the *x*-direction (*y* = 0). It shows that the temperature of the isotropic substrate is about 0.2 to 2 °C higher than that of the orthotropic substrate along the calculated path, and the corresponding temperature differences are both in a relatively small scale, 2.6 °C and 1.3 °C respectively, which reveals that the increase in the *y*-direction can manipulate the heat flux along the *y*-direction to reduce the maximum temperature, but it will slightly increase the uniformly of the temperature distribution along the *x*-direction. The above results are given by the theoretical analysis and FEA of mutual verification.

The quadrilateral network heater lattice (*χ*_0_ = 90°) is a square with a side length of 4 mm, as shown in Figure 4a. The theoretical temperature increase model for square topology is similar to the process above. Under the voltage applied at the left and right ends, the current density in the oblique path is 0 along the *y*-direction because of the vertical relationship between the two sides, as shown in Figure 4b. Therefore, only the line heat sources in the horizontal path along the *x*-direction have a power of 3.14 × 10^−3^ W, and the corresponding heat flux density is 0.5 W/m. Figure 4c shows the comparison between analytical and FEA results (see Appendix A, for details) of temperature distribution at the bottom of the isotropic and orthotropic substrate with a square network heater, respectively. For the isotropic substrate with *k_s_^x^* = *k_s_^y^* = *k_s_^z^* = 0.16 W·m^−1^·K^−1^, it can be seen that the temperature distribution of the substrate is not uniform, accompanying a temperature difference of around 5 °C. In order to improve the temperature distribution, we increase the thermal conductivity *k_s_^y^* to 1.6 W·m^−1^·K^−1^ with other parameters fixed. The temperature difference of the obtained orthotropic substrate is significantly reduced, which is no more than 2 °C as reflected from the temperature distribution contour map.

Figure 5 shows the temperature distributions along two paths of *x* = 0 and *y* = 0 of two different substrates (the origin is at the geometric center of the network heater lattice) to quantitatively evaluate the temperature homogenization effect of the orthotropic strategy on the square network heater heat transfer system. The temperature difference along the *y*-direction (*x* = 0) of the isotropic substrate is 5 °C, which is calculated from the lattice edge temperature of 4.7 °C and the center point temperature of 9.7 °C. Under the same conditions, the temperature difference of orthotropic is only 1.9 °C and the temperature distribution ranges from 6.2 to 8.1 °C along the path, as shown in Figure 5a. As discussed in the triangular heater topology, the increase in thermal conductivity *k_s_^y^* helps to obtain a more uniform temperature distribution along the *y*-direction. Figure 5b demonstrates the temperature of the isotropic substrate is 0.5 to 2 °C higher than that of the orthotropic substrate along the paths in the *x*-direction (*y* = 0). The corresponding temperature differences are 3.0 °C for the isotropic substrate and 1.9 °C for the orthotropic substrate, respectively. Consistent with the conclusion of triangular heater topology, the increase in the *y*-direction can manipulate the heat flux along the *y*-direction to reduce the maximum temperature. However, the difference is that it has little effect on the temperature uniformity along the *x*-direction.

The honeycomb network heater lattice (*χ*_0_ = 120°) is diamond shaped with a side length of 6.93 mm and two adjacent angles of 60° and 120°, as shown in Figure 6a. The theoretical temperature increase model for honeycomb topology is similar to the process above; only the linear superposition integral path needs to be changed to the function of the honeycomb lattice. Under the voltage applied at the left and right ends, the ratio of current density in the yellow, blue and gray path is 2:1:1 from the equivalent circuit approach combined with the periodic conditions (see Appendix A for details), as shown in Figure 6b. The powers of line heat sources in red, blue and gray paths are set as 3.14 × 10^−3^ W, 7.85 × 10^−34^ W and 7.85 × 10^−4^ W, respectively, and the corresponding heat flux densities are 0.5 W/m, 0.125 W/m and 0.125 W/m, respectively. Figure 6c shows the bottom temperature distribution comparison of the isotropic and orthotropic substrate with honeycomb network heater, which was calculated from both analytical and FEA (see Appendix A, for details). For an isotropic substrate with uniform thermal conductivity of 0.16 W·m^−1^·K^−1^, the temperature difference of the substrate bottom surface can reach around 7.6 °C. After increasing the thermal conductivity *k_s_^y^* to 1.6 W·m^−1^·K^−1^, the temperature difference drops to about 3 °C.

Figure 7 shows the temperature distributions along with *x* = 0 and *y* = 0 of two different substrates (the origin is at the geometric center of the network heater lattice) on the honeycomb network heater heat transfer system. The temperature difference along the *y*-direction (*x* = 0) of the isotropic substrate is 7.6 °C, which is calculated from the lattice edge temperature of 1.3 °C and the center point temperature of 8.9 °C. Under the same conditions, the temperature difference of orthotropic is 3.1 °C and temperature distribution ranges from 3.0 to 6.1 °C along the path, as shown in Figure 7a. As discussed in the triangular and square heater topology, the increase in thermal conductivity *k_s_^y^* helps obtain a more uniform temperature distribution along the *y*-direction. Figure 7b demonstrates the temperature distribution of isotropic and orthotropic substrates along the path in the *x*-direction (*y* = 0). The temperature difference along the *x*-direction of the isotropic substrate is 7.6 °C, which is calculated from the lattice edge temperature of 1.3 °C and the center point temperature of 8.9 °C. The temperature difference of orthotropic is 3.1 °C, and temperature distribution ranges from 3.0 to 6.1 °C along the path. Due to the high thermal conductivity in the y-direction, these regions (*y* coordinate within −3.5 to −1.5 and 1.5 to 3.5 in Figure 7a, *x* coordinate within −6 to −3 and 3 to 6 in Figure 7b) with low temperature are heated for a more uniform temperature distribution. Therefore, increasing the thermal conductivity in the *y*-direction can significantly achieve the ideal homogenization effect in both directions.

In order to systematically investigate the influence of the orthotropic substrate thermal conductivity on the temperature distribution uniformity of stretchable heat sources with different lattice configurations, the parametric calculation is carried out for the effects of the thermal conductivity ratios (*k_s_^y^/k_s_^x^*, *k_s_^z^/k_s_^x^* and *k_s_^x^/k_s_^z^*) on the temperature difference at the substrate bottom surface, as shown in Figure 8. The temperature difference of all these three types of network heaters decreases with the increase in *k_s_^y^/k_s_^x^*, and the temperature difference of the honeycomb configuration heater is less than that of the other two heaters, as shown in Figure 8a. This indicates that the effect of the orthotropic strategy to improve the thermal conductivity of the substrate in one direction on the homogenization of the temperature distribution of the honeycomb heat source is the weakest among the three heaters, which is consistent with the conclusion in Figure 7b. With the increase in *k_s_^z^/k_s_^x^*, the temperature difference of these three heaters increases slightly. When the ratio is large enough, i.e., the off-plane thermal conductivity is large enough, then the temperature distribution at the substrate bottom surface is almost the same as that of the network heaters. The temperature difference of the quadrilateral heat source is the largest, followed by the honeycomb, and the temperature difference of the triangle is the smallest, which is determined by the density distribution of heaters caused by different heater configurations, as shown in Figure 8b. The heat flux density of the quadrilateral heater is only distributed in the *x*-direction, and the spacing between heater network lines is large, so the temperature distribution difference is the largest. The heat flux distribution of the honeycomb heater is about twice the difference in the two directions, considering the large spacing between the heat source network lines, so the temperature distribution difference is in the middle. The distance between the heat flux distribution of the triangular heat source and the heater network lines are both the smallest of these three heaters, so the difference in temperature distribution is the smallest. Figure 8c demonstrates the effect of decreasing the thermal conductivity in the off-plane direction (i.e., the *z*-direction) on reducing the temperature distribution difference and enhancing the uniformity of temperature distribution. With the increase in *k_s_^x^/k_s_^z^*, the temperature difference at the bottom surface decreases, which means that the temperature distribution uniformity is enhanced. This indicates that decreasing the thermal conductivity in the *z*-direction is an alternative and effective strategy to enhance the uniformity of temperature distribution.

## 4. Conclusions

In summary, this paper presents an analytical orthotropic heat conduction model with three horseshoe lattice structures—triangular, square and honeycomb, which is verified by FEA. In order to improve the uniformity of the substrate temperature distribution, the orthotropic substrate is utilized to ameliorate the heat transfer system by increasing the thermal conductivity in one direction, while other parameters remain fixed. The temperature distribution at the substrate/skin interface is the most noteworthy of the EES heat transfer system, so we focus on the temperature on the substrate’s bottom surface. The results show that increasing the substrate thermal conductivity in the direction of heat flux spacing can effectively reduce the temperature difference of the substrate in this direction and increase the uniformity of temperature distribution. However, it has no obvious effect on the temperature uniformity in the other direction except for the honeycomb heater. In addition, increasing the thermal conductivity in the off-plane direction of the substrate will aggravate the heterogeneity of the substrate’s bottom surface temperature distribution. On the contrary, decreasing the thermal conductivity in the *z*-direction can significantly enhance the temperature uniformity of the substrate’s bottom surface. This study can be used to design the orthotropic substrate of stretchable hyperthermia electrodes, and it provides a theoretical basis for the mechanism and effect of temperature distribution regulation.

## Figures and Tables

**Figure 1 micromachines-13-01133-f001:**
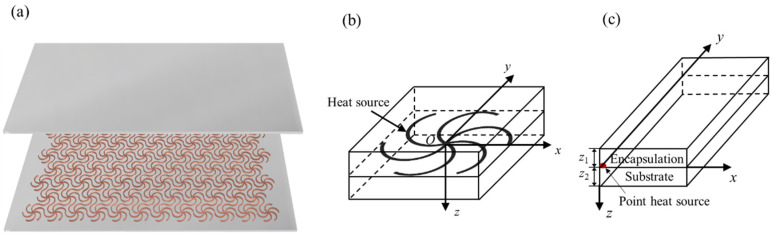
(**a**) Schematic diagram of the network heater based on a triangular lattice structure; (**b**) A single period of the network heater; (**c**) Schematic diagram of a point heater model.

**Figure 2 micromachines-13-01133-f002:**
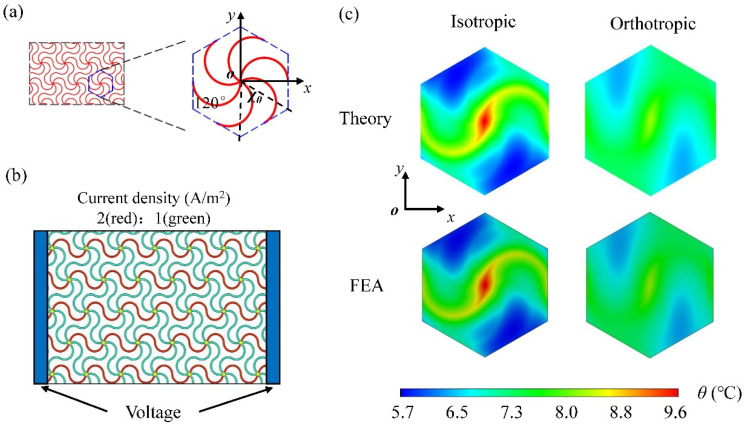
(**a**) Configurations of the stretchable triangular heat source; (**b**) Distribution of current density in the triangular heat source with the voltage applied at two ends; (**c**) Comparison of the theoretical and FEA temperature field on the bottom surface of the unit cell based on the isotropic and orthotropic substrate.

**Figure 3 micromachines-13-01133-f003:**
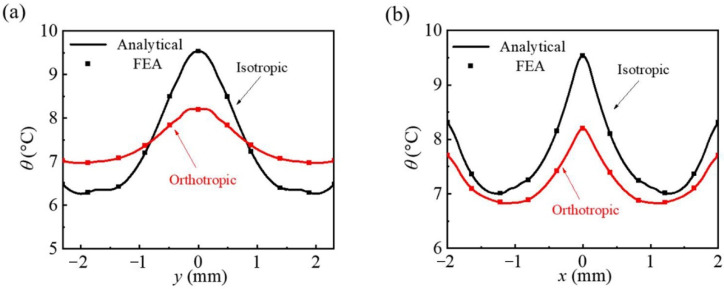
The comparisons of temperature increase along (**a**) *x* = 0 and (**b**) *y* = 0 on the bottom surface of the unit triangular heat source cell between the analytical results and FEA.

**Figure 4 micromachines-13-01133-f004:**
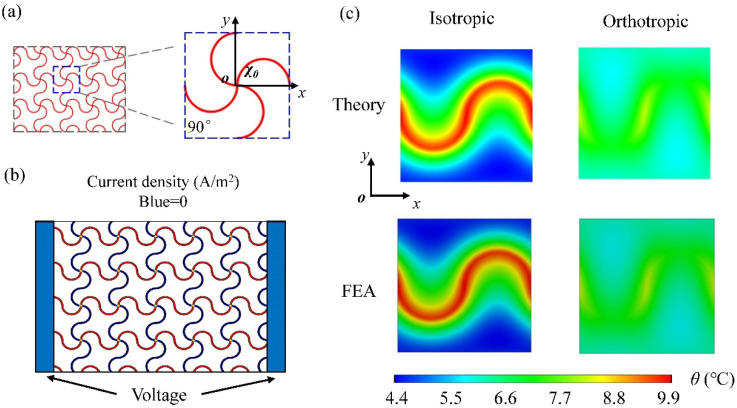
(**a**) Configurations of the stretchable square heat source; (**b**) Distribution of current density in the square heat source with the voltage applied at two ends; (**c**) Comparison of the theoretical and FEA temperature field on the bottom surface of the unit cell based on the isotropic and orthotropic substrate.

**Figure 5 micromachines-13-01133-f005:**
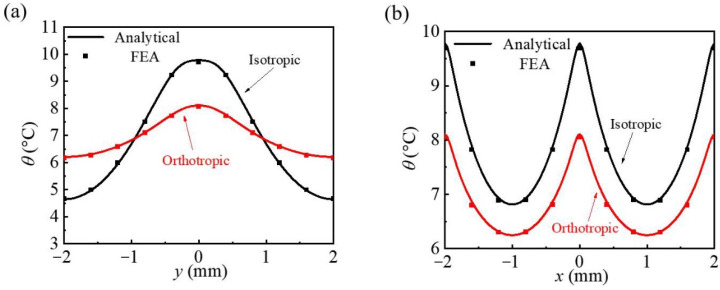
The comparisons of temperature increase along (**a**) *x* = 0 and (**b**) *y* = 0 on the bottom surface of the unit square heat source cell between the analytical results and FEA.

**Figure 6 micromachines-13-01133-f006:**
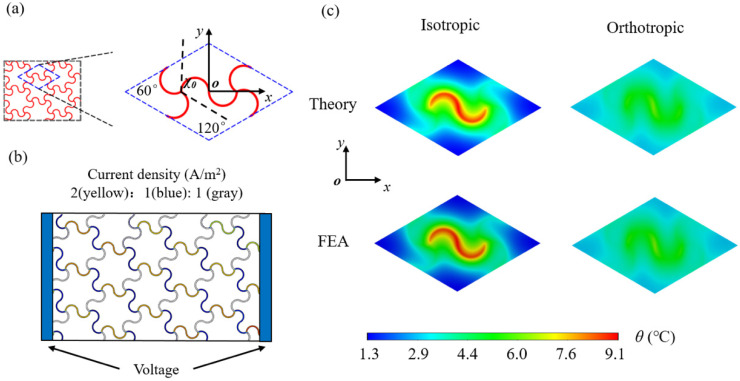
(**a**) Configurations of the stretchable honeycomb heat source; (**b**) Distribution of current density in the honeycomb heat source with the voltage applied at two ends; (**c**) Comparison of the theoretical and FEA temperature field on the bottom surface of the unit cell based on the isotropic and orthotropic substrate.

**Figure 7 micromachines-13-01133-f007:**
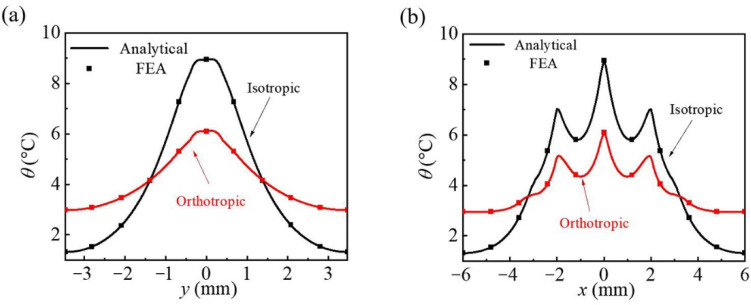
The comparisons of temperature increase along (**a**) *x* = 0 and (**b**) *y* = 0 on the bottom surface of the unit honeycomb heat source cell between the analytical results and FEA.

**Figure 8 micromachines-13-01133-f008:**
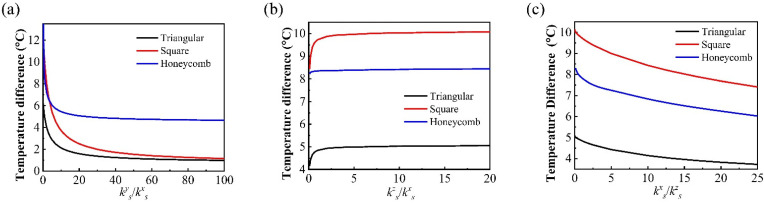
Effects of the orthotropic substrate with different ratios of (**a**) *k_s_^y^/k_s_^x^**,* (**b**) *k_s_^z^/k_s_^x^* and (**c**) *k_s_^x^/k_s_^z^* on the thermal uniformity of the bottom surface of the model.

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
