# Peer review of "An Analytic Orthotropic Heat Conduction Model for the Stretchable Network Heaters"

_micromachines, 2022, doi:10.3390/mi13071133_

Round 1

Reviewer 1 Report

In this work, the authors established heat conduction analytical models for three horseshoe lattice structures: triangular, square and honeycomb, then verified them by finite element analysis. The theoretical and simulation results are in good agreement, proving the accuracy of the model. In addition, the authors also confirmed that using a substrate with orthotropic thermal conductivity can significantly improve the uniformity of the temperature distribution of the heater at the substrate/skin interface, which is of great significance to the design and manufacture of wearable heaters.

However, some of the conclusions of this work appear to be wrong because the authors ignore the effect of adjacent unit cells in the finite element simulations of anisotropic substrates. The authors should redesign the simulation method and address this problem (see comments below for details). The reviewers consider this article acceptable for publication after addressing the following issues.

1.     For an orthotropic substrate, due to the dislocation arrangement of the unit cells, the minimum repeating unit is not a unit cell. For heat transfer simulations of orthotropic substrates, the simulation area is preferably rectangular or parallelogram and can cover the entire area with a linear array in the x/y direction (no misalignment or offset). For the anisotropic case in this work (higher thermal conductivity in the y-axis direction), the left and right boundaries of the simulation area should be perpendicular to the x-axis. Especially for the case of Figure 6, the authors can take a rectangular unit cell for simulation. The rectangle sides should be parallel to the x and y axes.

2.     The statement “In areas far away from high heat flux, this strategy has little effect on temperature. ...... For the honeycomb heat source, the lines with high heat flux density are distributed discretely in both directions, as shown in Fig. 6b”. This conclusion seems wrong. As mentioned above, the discussion here does not take into account the heat conduction from the obliquely adjacent unit cells to the edges of this unit cell (x coordinate within -6 to-3 and 3 to 6 in Figure 7b). In fact, the x-coordinate of the main heating region of the adjacent unit cell just covers the low temperature region in Figure 7b. Due to the high thermal conductivity in the y-direction, these regions will be heated for a more uniform temperature distribution.

3.     If the journal allows adding supplementary information, it's better to provide a simple calculation for the current density ratio in different paths for three unit cells in the supplementary information to facilitate the understanding of general readers.

4.     If the journal allows adding supplementary information, it's better to give the meshing of 3D models and 3D simulation results (e.g. layered slice temperature fields) in Supplementary Information.

5.     It is better to add the x/y axis in Figure 2a to facilitate the understanding of readers.

6.     The statement “Fig 5b demonstrates the temperature of the isotropic substrate is 0.5 to 2 higher than that of the orthotropic substrate along the paths in the x-direction (y = 0). The corresponding temperature differences are 2.0 ℃ for isotropic substrate”, but in Figure 5b it’s 3 .

7.     In Figure 6a, blue and green colors are similar and easy to confuse, it is better to change them with a higher contrast.

8.     The statement “After increasing the thermal conductivity ksy to 1.6 W/m/K, the temperature difference drops to about 2 ”. According to Figure 7b, the temperature difference in the x-axis direction of the anisotropic substrate is greater than 4 . In fact the temperature difference will indeed drop to around 2 after the authors use the correct simulation area.

9.     Can increasing both ksx and ksy, or decreasing ksz, provide a sufficiently uniform temperature distribution in the x-y plane and reduce temperature distribution differences on the substrate bottom surface? If possible, it would be better to add a plot with ksx/ksz as the abscissa to illustrate this (ksx=ksy).

10.  This paper proposes that orthotropic substrates can promote uniform thermal distribution. The authors can briefly introduce the mainstream methods for preparing anisotropic substrates in the Introduction section.

Reviewer 2 Report

This manuscript describes an analytical orthotropic heat conduction model with three horseshoe lattice structures of triangular, square, and honeycomb, which is verified by finite element analysis (FEA). Overall, this work is characterized through well-established analytical modelling. Although this work looks interesting, the authors are asked to significantly improve the manuscript for publication. More specific concerns and suggestions are given below.

1. The authors characterized the orthotropic heat conduction with three horseshoe lattice structures. It would be recommended to characterize the orthotropic heat with network heat source in stretched state since the underlying application is epidermal electronics.

2. The authors should improve both language and organization quality.

3. In (Page 6, line 4), (page 6, line 25), (page 6, line 33), (page 8, line 9), (page 8, line 12), (page 9, line 27), the authors write W/m/K as the unit of thermal conductivity. It needs to be corrected to W/m·K or W·m-1·K-1.

4. Page 6, line 5, W/m2/K needs to be corrected to W/m2·K or W·m-2·K-1

5. It is advisable to provide detailed information such as the size, and the dimension of triangular lattice, square lattice, and honeycomb lattice, respectively..

6.  Page 2, line 10, grapheme should be graphene.

Reviewer 3 Report

The models of orthotropic heat conduction are interesting but the current densities of the models are different. 

The results of the efficiency of heat are needed with power when we compared the heat conduction models

Reviewer 4 Report

In this paper, the authors carried out a temperature distribution manipulation strategy for orthotropic substrates with an aim to improve the temperature uniformity of the stretchable heater constructed with 2D serpentine structured electrodes. In specific, the effect of orthotropic substrate on the uniformity of the temperature distribution in three types (i.e., triangular, square, and honeycomb) of heat transfer systems is investigated. Considering the parametric study with systematic analysis, I agree that the work is suitable for publication in Micromachines after reasonable responses to the questions listed below.

1. In general, heat conduction is a function of time, and hence the temperature of a body varies with time. However, the authors did not mention such transient heat conduction with respect to time in the manuscript. The temperature variation as presented in Figures 3, 5, and 7 may be even bigger or smaller depending on the time of the interest, particularly with the low heat conductivity of the Ecoflex elastomer. Thus, it is suggested to present the temperature variation with respect to time and clarify this issue accordingly in the revised manuscript.

2. The authors mentioned in the manuscript on Page 6, “the substrate will aggravate the heterogeneity of the substrate bottom surface temperature distribution.”. How significant is it to provide a temperature difference of ~5 oC for the biophysical therapy heaters in EES? It is understandable that the temperature variation can be reduced depending on the structure type and orthotropic substrates. However, the size of the unit structure can be manufactured in the order of a few hundreds of microns or even smaller, and hence the reviewer is curious about how significant it would be to provide an orthotropic substrate which will be another burden of the manufacturing and increase in the cost. The authors should clarify this issue in the revised manuscript.

Round 2

Reviewer 1 Report

The authors fully responded to the reviewer's comments and addressed the issues raised by the reviewer. The authors modified the simulation model and improved the analysis and discussion of the simulation results. For the reviewer, the article can be accepted after addressing the following minor issues:

1.      In Response 7, the authors changed the colors of the lines in Figure 6a to yellow, blue and gray. But in main text it’s still red, blue and gray, see the statement “Under the voltage applied at the left and right ends, the ratio of current density in the red, blue and gray path is 9: 5: 4”.

2.      Still this statement “Under the voltage applied at the left and right ends, the ratio of current density in the red, blue and gray path is 9: 5: 4”. However, the current ratio I calculated here is 2:1:1. Maybe my calculation is wrong, could the authors please also put the calculation of current density ratio of the honeycomb heat source in the supplementary information? Thank you.

3.      It seems that Ref 31 is cited before Refs 29 and 30 in the Introduction.

Reviewer 2 Report

As I commented, the manuscript can be improved if the authors characterize the orthotropic heat with network heat source in stretched state. However, I understand that it would be hard and they will be continuously exploring it. Thus, I recommend this manuscript to be published.
